# Safety and Discontinuation Rate of Dimethyl Fumarate (Zadiva^®^) in Patients with Multiple Sclerosis: An Observational Retrospective Study

**DOI:** 10.3390/jcm12154937

**Published:** 2023-07-27

**Authors:** Roya Abolfazli, Mohammad Ali Sahraian, Atefeh Tayebi, Hamidreza Kafi, Sara Samadzadeh

**Affiliations:** 1Department of Neurology, Amiralam Hospital, Tehran University of Medical Sciences, Tehran 11457-65111, Iran; 2Multiple Sclerosis Research Center, Neuroscience Institute, Tehran University of Medical Sciences, Tehran 19978-66837, Iran; sahraian1350@yahoo.com; 3Food Industry Engineering, Tehran Islamic Azad University of Medical Sciences, Tehran 19395-1495, Iran; atefehtayebi2012@yahoo.com; 4Department of Medical, Orchid Pharmed Company, Tehran 19947-66411, Iran; kafi.h@orchidpharmed.com; 5Charité—Universitätsmedizin Berlin, Corporate Member of Freie Universität Berlin and Humboldt-Unverstät zu Berlin, Experimental and Clinical Research Center, 13125 Berlin, Germany; 6Department of Regional Health Research and Molecular Medicine, University of Southern Denmark, 5230 Odense, Denmark

**Keywords:** dimethyl fumarate, Zadiva^®^, multiple sclerosis, real-world practice, safety, tolerability

## Abstract

Background: This study evaluates the real-world safety and discontinuation rate of Zadiva^®^ (generic product of dimethyl fumarate (DMF)) in Iranian patients with relapsing–remitting multiple sclerosis (RRMS), supplementing existing clinical evidence from randomized controlled trials. Methods: This retrospective observational study evaluated the real-world safety and discontinuation rate of DMF in RRMS patients from Amir A’lam referral hospital’s neurology clinic. Data on safety, discontinuation rate, and clinical disease activity were collected retrospectively. The study aimed to assess the discontinuation rate, safety, and reasons for discontinuation, as well as the number of patients experiencing a relapse, MRI activity, and EDSS scores. Results: In total, 142 RRMS patients receiving DMF were included in the study, with 15 discontinuing treatment due to adverse events, lack of efficacy, or pregnancy. Notably, a significant reduction in relapse rates was observed, with 90.8% of patients remaining relapse-free throughout the study period. After 1 year of treatment with Zadiva^®^, only 17.6% of patients experienced MRI activity, whereas the EDSS score remained stable. Conclusions: This study provides important real-world data on the safety and tolerability of Zadiva^®^ in RRMS patients. The results indicate that Zadiva^®^ is generally well tolerated and safe, with a low discontinuation rate due to adverse events or lack of efficacy. These findings suggest that Zadiva^®^ is an effective and safe treatment option for RRMS patients in real-world practice.

## 1. Introduction

Multiple sclerosis (MS) is a chronic and multifaceted autoimmune disorder marked by the involvement of immune-mediated inflammation, demyelination, and axonal degeneration in the central nervous system (CNS). These pathological hallmarks of the disease can manifest in a range of neurological symptoms and functional impairments that ultimately contribute to significant morbidity and disability in affected individuals [1]. According to epidemiological studies, the prevalence of MS in Iran is 100 per 100,000 individuals, indicating a high burden of the disease. Furthermore, the prevalence is markedly higher in certain provinces [2]. Considering the high burden of the disease in Iran, evaluating the safety and efficacy of medications used for MS treatment is crucial.

In recent years, the availability of numerous disease-modifying therapies (DMTs) has provided new options for managing MS, with the approval of oral DMTs offering greater convenience and improved therapeutic compliance for patients [3,4]. One such therapy is dimethyl fumarate (DMF), an oral immunomodulatory DMT approved for reducing disease activity in patients with relapsing–remitting multiple sclerosis (RRMS). The clinical efficacy of DMF, as evaluated through annualized relapse rate, disability worsening, and radiological activity, was demonstrated in two phase III randomized clinical trials (CONFIRM and DEFINE), leading to its approval in 2013 [5,6]. The favorable long-term efficacy and safety profile of DMF, as reported in an open-label extension study, has resulted in an increase in its prescribing following its approval [7,8].

Randomized controlled trials typically provide high-quality evidence of drug efficacy under controlled and ideal conditions. However, these trials may not always reflect the complex and diverse real-world contexts in which medications are used. Thus, post-marketing observational studies can provide additional insights into the safety and tolerability of therapy in the broader patient population. The primary objective of this retrospective cohort study was to assess the safety and tolerability of Zadiva^®^ (a brand-generic product of DMF manufactured by NanoAlvand Co., Tehran, Iran) in Iranian patients with RRMS under real-world clinical practice conditions. By examining the drug’s safety and tolerability in various patient populations and clinical scenarios, the study aimed to provide a more comprehensive understanding of the drug’s safety profile. This information is essential for informed decision-making and optimal patient management, ultimately leading to improved clinical outcomes for patients with RRMS.

## 2. Materials and Methods

### 2.1. Study Design, Ethical Considerations, and Objectives

This real-world observational, retrospective study was conducted in accordance with the Declaration of Helsinki and approved by the Ethics Committee of Amir A’lam University Hospital. Informed consent was obtained from all patients included in this real-world study. The primary objectives were to evaluate safety, discontinuation rates during follow-up, and reasons for discontinuation. Additionally, the study aimed to determine the number of patients who experienced a relapse, MRI activity, and the EDSS score.

### 2.2. Study Population

In total, 142 patients were included in the study, of whom 126 (88.73%) were female. All patients had a confirmed diagnosis of MS according to the revised McDonald criteria 2017 [9] and were referred to the neurology clinic of Amir A’lam University Hospital (Tehran, Iran) between October 2020 and October 2021. Patients who were prescribed Zadiva^®^ for MS were selected. Patients were ambulatory with an EDSS score between 0 to 5.5 at the time of starting Zadiva^®^. At the time of analysis, all patients had received Zadiva^®^ treatment for at least one year. During this first year, the safety, discontinuation rate, and effectiveness of the treatment were all evaluated.

### 2.3. Treatment Protocol

DMF was administered using a gradually increasing dosing regimen over four weeks to improve gastrointestinal tolerability. The regimen consisted of a starting dose of 120 mg once daily (in the evening) for seven days, followed by 120 mg two times daily (morning and evening) for seven days, then 120 mg of DMF in the morning and 240 mg in the evening for seven days, and finally, 240 mg DMF twice daily (morning and evening) from the fourth week until the end of the study period.

The gradual dose escalation over four weeks was employed at Amir A’lam Hospital to enhance gastrointestinal tolerability and improve patient compliance with the medication. By gradually increasing the dosage, the aim was to minimize the likelihood of side effects, especially gastrointestinal issues and flushing, which are commonly associated with DMF. This approach was chosen based on clinical experience and considerations of optimizing patient tolerability and adherence to the medication [10,11].

It is important to note that prior to initiating treatment with DMF, patients underwent the appropriate washout period based on the specific DMT previously administered. Patients transitioning from interferon therapies or glatiramer acetate did not require a washout period due to the relatively short half-lives of these medications. However, for patients switching from fingolimod or teriflunomide, a washout period of approximately 4 weeks was observed to ensure adequate separation between the previous DMTs and the initiation of DMF treatment.

### 2.4. Data Collection

After obtaining informed consent from the participants, data on clinical disease activity and safety were retrospectively collected from the patients’ medical records. This included patients’ demographic and baseline characteristics, MS history (disease diagnosis date, relapse history, and the expanded disability status scale (EDSS) score before starting DMF), prior disease-modifying therapies (DMTs), and the start date of DMF. Data regarding the treatment’s effectiveness included relapses, magnetic resonance imaging (MRI) activity, and EDSS score, all evaluated over the course of the first year of treatment. A relapse was defined as a monophasic clinical episode with patient-reported symptoms and objective findings typical of MS, reflecting a focal or multifocal inflammatory demyelinating event in the CNS, developing acutely or subacutely, with a duration of at least 24 h, with or without recovery, and in the absence of fever or infection [9]. Additionally, MRI activity was defined as the detection of any new/enlarging T2 or gadolinium-enhancing lesions within this one-year period.

Adverse Events Assessment:

Medical records were utilized to collect all adverse events (AEs) observed during the first year of treatment in this real-world study. All AEs were classified based on the Medical Dictionary for Regulatory Activities (MedDRA) as the preferred term (PT) and system organ class (SOC). Serious adverse events (SAEs) were documented according to the ICH guideline (E2B). An SAE was defined as any AE that “results in death, is life-threatening, requires in-patient hospitalization or prolongation of existing hospitalization, results in persistent or significant disability or incapacity, results in congenital anomaly or birth defect.” Occurrences of gastrointestinal disorders, including diarrhea, dyspepsia, nausea, and vomiting, were assessed.

### 2.5. Data Analysis

It should be noted that the sample size was not estimated based on statistical power since this was a real-world study conducted on available patients. All statistical analyses were performed using RStudio version 4.3.1 (RStudio, Inc., Boston, MA, USA) with relevant packages, including stats, ggplot2, tidyverse, and exact 2 × 2 for executing the McNemar test. Descriptive statistics, including mean and standard deviation for continuous variables and frequency and percentage for categorical variables, were used to summarize the data. McNemar’s test was used to compare the percentages of relapse-free patients one year before and after starting DMF.

Safety evaluation was reported based on the incidence rate of adverse events. The incidence of AEs was summarized according to system organ class and preferred term of AEs. Additionally, we have reported the number of patients who did not experience any adverse events during this first one-year treatment period.

## 3. Results

### 3.1. Demographics and Baseline Characteristics

All of the demographic and baseline characteristic data are demonstrated in Table 1. In total, 142 patients with MS were included in this retrospective study. The mean age of the patients was 33.90 ± 7.76 years, and the mean EDSS score at inclusion was 1.64 ± 0.44. Most of the patients were female, and 83.8% had received at least one disease-modifying therapy (DMT) prior to DMF initiation. Interferons were the most prescribed DMT used before DMF initiation. As shown in the table, the mean disease duration was 4.55 ± 3.82 years.

### 3.2. Safety Profile

Table 2 provides an overview of the key safety results, presenting the incidence of all adverse events by preferred term (PT) and system organ class (SOC). Out of 142 enrolled patients, 140 adverse events were recorded and no serious adverse events (SAEs) were reported. Importantly, 81 patients experienced no adverse events during the course of the study.

The most commonly reported adverse events were flushing (23.94%), abdominal pain (16.2%), increased transaminases (11.27%), and diarrhea (9.15%). These findings highlight the overall tolerability of the medication in our patient population. It is important to note that the dosage regimen used in this study (as described earlier) played a role in managing the occurrence and severity of adverse events.

### 3.3. Discontinuation Rate

Of the 142 patients included in the study, 127 (89.43%) remained on DMF at the end of the data collection period, whereas 15 (10.56%) discontinued the treatment due to various reasons. Adverse events, lack of efficacy, and pregnancy were the most common reasons for discontinuation (Table 3). Of the patients who discontinued treatment, 8 out of 15 (53.33%) did so due to AEs, with GI disorders being the most common (5/15, 33.33%), followed by skin complications (2/15, 13.33%) and lymphopenia (1/15, 6.66%). Additionally, five patients discontinued treatment due to treatment failure, all of whom had MRI activity and one of whom also experienced a relapse. Following discontinuation due to lack of efficacy, two patients switched to ocrelizumab and one patient switched to rituximab.

### 3.4. Effectiveness Outcomes

All enrolled patients received DMF following a confirmed relapse. As shown in Table 4, after applying McNemar’s test, we found a significant reduction in the number of patients with relapses after initiating DMF treatment. After treatment, 129 out of 142 patients (90.85%) experienced no relapses. This change was highly significant, with a *p*-value of less than 0.001. Moreover, the odds of patients having a relapse after treatment were approximately 10 times higher before treatment, with a 95% confidence interval ranging from 5.60 to 19.14. Out of 142 patients, 25 (17.61%) had MRI activity after 1 year of DMF treatment. Among these patients, 18 (72%) had 1 new or enlarging T2 lesion, 2 (8%) had 2 new or enlarging T2 lesions, 3 (12%) had 1 gadolinium-enhancing lesion, 1 (4%) had 2 gadolinium-enhancing lesions in addition to 1 new or enlarging T2 lesion, and 1 (4%) had 1 gadolinium-enhancing lesion and 2 new or enlarging T2 lesions.

Table 5 displays that the mean EDSS score for the patients increased slightly from 1.64 ± 0.44 before starting DMF to 1.68 ± 0.50 after one year of DMF treatment. However, the difference is not statistically significant.

## 4. Discussion

### 4.1. Lower Incidence of GI Adverse Events Attributed to Slow Dose Titration of DMF

The present study adds to the existing literature regarding the safety and tolerability profile of dimethyl fumarate in RRMS patients in a real-world setting. The lower incidence of GI adverse events observed in our study compared to previous studies suggests that a slow dose titration of DMF, as implemented in our study, may be beneficial in reducing the incidence of GI disorders. This finding is consistent with a post-marketing study by Alroughani et al., who reported GI disorders as the most common adverse event in 134 RRMS patients receiving DMF [12]. However, a retrospective study by Vollmer et al. reported GI disorders in 80.5% of patients [13]. Similarly, a single-center study by Barros et al. reported a 42% incidence of GI disorders, with abdominal pain, diarrhea, nausea, and vomiting occurring in 26.7%, 14.8%, 9.6%, and 5.7% of patients, respectively [14,15]. Our study highlights the importance of monitoring adverse events in patients receiving DMF and the potential benefits of a slow dose titration to improve the tolerability of this treatment. Further studies are needed to confirm these findings and evaluate the optimal dosing strategy for DMF in RRMS patients.

Furthermore, our study found a lower incidence of flushing compared to previous observational studies by Rodríguez-Regal et al. and Conde et al., where flushing was reported in 43.3% and 48% of patients, respectively [16,17]. Administering the medication with food may reduce the incidence of flushing, as reported in a study by Kappos et al. [18].

### 4.2. Lower Discontinuation Rate of DMF in Real-World Settings

As previously mentioned, our study had a low rate of discontinuation of DMF, with only 15 out of 142 patients (10.56%) discontinuing the treatment. Of these, 8 out of 142 patients (5.63%) discontinued due to adverse events and 5 out of 142 patients (3.52%) discontinued due to lack of efficacy. In comparison, a study conducted by Rodríguez-Regal et al. on 90 patients reported that 12 patients (13.5%) discontinued treatment, with 3 patients (3.4%) discontinuing due to therapeutic failure and 8 patients (8.9%) discontinuing due to adverse events [16].

Another study by Hersh et al. on 337 RRMS patients reported a higher rate of discontinuation during a 12-month follow-up period, with 100 patients (29.7%) discontinuing dimethyl fumarate treatment. Of these, 28 patients discontinued due to disease activity, including 9 cases of clinical relapse, and 73 patients discontinued due to intolerability [19]. The results of a 24-month follow-up period by the same study demonstrated that of the 293 RRMS patients who received dimethyl fumarate, 127 (43.3%) discontinued treatment. Of these patients, 30 (10.2%) discontinued due to disease activity, including 7 cases of clinical relapse, and 76 patients (25.9%) discontinued due to adverse events [20].

On the other hand, a study by Zecca et al. reported a discontinuation rate of 28.5%, with 17.7% of patients discontinuing due to adverse events and 7% discontinuing due to insufficient effectiveness [21]. Similarly, a French retrospective study by Conde et al. reported a discontinuation rate of 18.3%, with 27.5% of patients discontinuing due to adverse events. Our study findings are consistent with other studies that reported adverse events, particularly GI disorders, as a major factor in discontinuing treatment with DMF [17]. However, some real-world studies reported a higher discontinuation rate than ours due to a lack of efficacy [13,22,23,24]. Further studies with larger sample sizes and longer follow-up periods are needed to confirm these findings and evaluate the long-term safety and efficacy of DMF in the management of RRMS.

### 4.3. Relapse-Free Rates Post Treatment with DMF

Based on the findings of our study, 90.85% of the patients treated with DMF for one year were found to be relapse-free during that year. This result is in line with the findings of previous independent studies, such as the multicenter post-marketing study by Prosperini et al., which reported a 90% relapse-free rate in 275 patients treated with dimethyl fumarate during 12 months of follow-up [8]. Zecca et al. also reported an 84.8% relapse-free rate in their one-year post-marketing survey involving 134 patients [21]. Furthermore, Berger et al. conducted a phase IV, open-label, single-arm, observational, multicenter study in Europe and Canada and reported that 88% of the patients remained relapse-free 12 months after initiating dimethyl fumarate treatment [25]. These findings provide further evidence of the effectiveness of DMF in reducing the risk of relapses in patients with relapsing–remitting multiple sclerosis.

### 4.4. Effectiveness of DMF in the Treatment of RRMS

The effectiveness of DMF in RRMS treatment has been evaluated in various studies. Our study found that 90.85% of patients were relapse-free post treatment with DMF, consistent with previous studies by Prosperini et al., Zecca et al., and Berger et al., who reported that 90%, 84.8%, and 88% of patients, respectively, were relapse-free during 12 months of follow-up [21,25,26]. Furthermore, several retrospective studies by Mirabella et al., Vollmer et al., and Rodríguez-Regal et al. have also supported our findings, reporting low relapse rates or high relapse-free rates in patients treated with dimethyl fumarate.

In addition to assessing relapse rates, Mirabella et al. reported that the mean EDSS score progression in Italy was 0.08 ± 0.44 per year in their independent, multicenter, retrospective post-marketing study [27]. However, other real-world studies reported minor improvements in EDSS scores. Our study found that the EDSS score remained stable [21,25].

### 4.5. Limitation

Our study was not without limitations. One major limitation was the potential for bias due to missing records in retrospective cohort studies. Additionally, adverse events may have been under-reported, leading to a potentially inaccurate estimation of their true incidence. Furthermore, the study did not assess the incidence of rare adverse events due to the sample size not being calculated based on statistical power or rare event incidence. Another limitation is the absence of a control group in our study design. Although each patient served as the control to some degree, by comparing their post-treatment condition with their pre-treatment baseline, the lack of an external control group may limit the generalizability of our findings. One notable advantage of the study, however, was that all patients were evaluated by a single physician, which helped ensure consistency in the assessments.

## 5. Conclusions

The present retrospective study confirms the safety and tolerability of Zadiva^®^ in a real-world setting for one year, supporting its potential as a safe and effective treatment option for MS patients who have experienced a relapse. The lower incidence of GI adverse events in our study suggests that a slower dose titration of DMF may help reduce the incidence of GI disorders. The findings of this study are consistent with other real-world studies, highlighting the importance of monitoring adverse events in MS patients. No serious adverse events were reported, and there was a significant reduction in the number of patients with relapses, supporting the use of Zadiva^®^ in the management of RRMS. However, the limitations of the study should be taken into account, and further studies with larger sample sizes and longer follow-up periods are required to confirm the results and assess the risk of rare adverse event incidence. Nonetheless, this study contributes to the growing body of evidence supporting the safety and efficacy of Zadiva^®^ and its potential as a promising therapy for MS patients.

## Figures and Tables

**Table 1 jcm-12-04937-t001:** Baseline characteristics of study participants with relapsing–remitting multiple sclerosis.

Characteristic	Value
Age (years)	33.90 ± 7.76 (Mean ± SD)
Gender (female), *n* (%)	126 (88.73)
EDSS score	1.64 ± 0.44 (Mean ± SD)
Duration of disease (years) *	4.55 ± 3.82 (Mean ± SD)
Previous medication history, *n* (%)	
Interferon beta-1a, IM	50 (35.21)
Interferon beta-1a, SC	21 (14.79)
Interferon beta-1b, SC	9 (6.34)
Glatiramer acetate	36 (25.35)
Fingolimod	2 (1.41)
Teriflunomide	1 (0.70)
Naive	11 (7.75)
Unknown	12 (8.45)

Note: Values are presented as mean ± standard deviation (SD) or number and percentage (*n* (%)) of patients. * Calculated for 139 patients.

**Table 2 jcm-12-04937-t002:** Incidence of adverse events (AEs) classified by preferred term (PT) and system organ class (SOC).

System Organ Class	Female/Male Ratio (Total Events)	Preferred Term Name	No. of Patients (*n* = 142)
Vascular disorders	34:0	At least one event	34 (23.94%)
		Flushing	34 (23.94%)
Gastrointestinal disorders	48:10	At least one event	27 (19.01%)
		Abdominal pain	23 (16.2%)
		Diarrhea	13 (9.15%)
		Dyspepsia	11 (7.75%)
		Nausea	9 (6.34%)
		Vomiting	2 (1.41%)
Skin and subcutaneous tissue disorders	18:1	At least one event	17 (11.97%)
		Erythema	10 (7.04%)
		Rash	4 (2.82%)
		Pruritus	4 (2.82%)
		Lichenoid keratosis	1 (0.7%)
Investigations	12:4	At least one event	16 (11.27%)
		Transaminases increased	16 (11.27%)
Blood and lymphatic system disorders	6:1	At least one event	7 (4.93%)
		Lymphopenia	6 (4.23%)
		Eosinophilia	1 (0.7%)
Infections and infestations	3:0	At least one event	3 (2.11%)
		Varicella zoster virus infection	2 (1.41%)
		Tinea versicolor	1 (0.7%)
Nervous system disorders	2:0	At least one event	2 (1.41%)
		Headache	2 (1.41%)
Immune system disorders	1:0	At least one event	1 (0.7%)
		Urticaria	1 (0.7%)

**Table 3 jcm-12-04937-t003:** Reasons for discontinuation of DMF treatment in patients with RRMS.

Reason for Discontinuation	Patients (Number, Percentage)
Pregnancy	2 (1.41%)
Lack of efficacy	5 (3.52%)
GI Side effects	5 (3.52%)
Lymphopenia	1 (0.7%)
Skin complications	2 (1.41%)

**Table 4 jcm-12-04937-t004:** Comparison of number of relapses before and after treatment with DMF.

Number of Relapses	Post-Treatment (*n*, %)	Pre-Treatment (*n*, %)	*p*-Value	Odds Ratio	95% Confidence Interval
0	129 (90.85%)	0 (0.0%)	<0.001 (<2.2 × 10^−16^)	9.92	5.60–19.14
1	13 (9.15%)	142 (100%)			

The *p*-value, odds ratio, and 95% confidence interval were determined using McNemar’s test, comparing the paired nominal data before and after treatment.

**Table 5 jcm-12-04937-t005:** Mean EDSS score before and after treatment with DMF.

Treatment Phase	EDSS (Mean ± SD)	95% CI	Mean Difference (95% CI)	*p*-Value
Pre-treatment	1.64 ± 0.44	(1.57, 1.71)	-	-
Post-treatment	1.68 ± 0.50	(1.60, 1.77)	−0.05 (−0.177, 0.077)	0.1257

A paired *t*-test was performed to compare the scores, yielding a non-significant result (t = −5, df = 1, *p* = 0.126).

## Data Availability

The datasets generated and/or analyzed during the current study are not publicly available but are available from the corresponding author upon reasonable request.

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
