# Peer review of "Safety and Discontinuation Rate of Dimethyl Fumarate (Zadiva®) in Patients with Multiple Sclerosis: An Observational Retrospective Study"

_jcm, 2023, doi:10.3390/jcm12154937_

Round 1
Reviewer 1 Report
The manuscript entitled "Safety and Discontinuation Rate of Dimethyl Fumarate (Zadiva®) in Patients with Multiple Sclerosis: An Observational Retrospective Study" is an original article describing the real-world evidence of dimethyl-fumarate use in a group of 142 patients with multiple sclerosis, in a retrospective study. There are some issues that require correction:
- in Material and Methods, line 82 - the statement lacks subject
- there should be a definition of a relapse included in Methods
- line 159 - the statement "two patients switched to ocrelizumab" has been repeated
- line 272 - "a low incidence of SAEs" should be changed to "no SAEs" (like in line 145)
- drug name "Zadiva" should be rather used after name of the substance, e.g. DMF (Zadiva).
Generally, the studied group was rather small and the observation period was quite short - only one year. The article requires also English language editing.
English language should be improved by a native speaker.
Author Response
The manuscript entitled "Safety and Discontinuation Rate of Dimethyl Fumarate (Zadiva®) in Patients with Multiple Sclerosis: An Observational Retrospective Study" is an original article describing the real-world evidence of dimethyl-fumarate use in a group of 142 patients with multiple sclerosis, in a retrospective study. There are some issues that require correction:
- there should be a definition of a relapse included in Methods
|
Thank you so much for your constructive comments. We have carefully revised the manuscript accordingly.
The text is now revised.
The definition of a relapse is now incorporated into the Methods section based on the McDonald Criteria 2017. A relapse was defined as a monophasic clinical episode with patient-reported symptoms and objective findings typical of MS, reflecting a focal or multifocal inflammatory demyelinating event in the CNS, developing acutely or subacutely, with a duration of at least 24 hours, with or without recovery, and in the absence of fever or infection
The text is now revised.
The text is now revised.
Thank you for your valuable feedback. We apologize for the oversight and agree with your suggestion to introduce the drug as DMF (Zadiva) for better clarity. We have now revised the manuscript accordingly and replaced "Zadiva" with "DMF (Zadiva)" throughout the document, except for the conclusion and the initial part of the methods section where the use of the brand name seemed more appropriate. We appreciate your attention to detail, and this change will surely enhance the understanding of our study for readers.
We appreciate your insightful comments. We understand your concern regarding the size of the study group and the duration of the observation period. The current study, being an initial exploration, was designed to examine the effects of DMF (Zadiva) over a one-year period. We acknowledge that a larger sample size and longer observation period could potentially yield more comprehensive results. We aim to extend our research in future studies to address these points.
As for the English language editing, we have now strived to improve the English further. |

Reviewer 2 Report
This manuscript investigates the safety and effectiveness of dimethyl fumarate as a treatment for relapsing-remitting multiple sclerosis over the course of one year in a patient population in Iran in a retrospective cohort study. This manuscript highlights important safety data and suggests an even more gradually increasing dosing regimen to reduce the number of adverse effects. There are several issues that warrant the editor’s attention listed below.
Several authors have affiliations with a pharmaceutical company, including one author with their only affiliation listed as the same pharmaceutical company. Please clarify whether there is any conflict of interest or if this pharmaceutical company is involved with Zadiva® or NanoAlvand Co.
Further clarification would be appreciated with regard to the Author Contributions, where V.G. is listed as the sole author involved in the investigation. According to author affiliations, V.G. is affiliated with the Department of Business of a pharmaceutical company as well as with the faculty of veterinary medicine of the Tabriz Branch of the Islamic Azad University.
Please attach the informed consent form as a supplement.
The length of the study period is not included in the Materials and Methods section. There are several references in the statistical analyses in the Results section referring to differences after one year of treatment with Zadiva®, but there is no explicit mention of the study length. Please include this information in either the Treatment Protocol subsection or any other appropriate subsection.
Additionally, the background information does not include any information regarding the expected length of time between relapses in patients not treated with Zadiva® at any point in the paper. This information would be important to make statements regarding the efficacy of the medication as one of the claims of the paper is a significant reduction in relapses (Lines 37, 164, and 271).
Please include the breakdown of male and female patients in the Study Population subsection within the Materials and Methods section and discuss critically how this might have impacted the results.
Please provide additional explanation for the rationale supporting the decision to utilize a different treatment regimen (120mg OD to 120mg BID to 120mg OD + 240mg OD to 240mg BID) from the treatment regimen in the medication insert of Zadiva® (which is the same as the standard treatment regimen for generic dimethyl fumarate).
The Results section includes a table of previous DMTs patients had received prior to Zavida® initiation, but there is no mention of whether these patients stopped their previous DMT prior to beginning treatment with Zavida®, and if so, how long they stopped prior to initiating the Zavida® treatment. Please clarify and discuss how this might have impacted the therapeutical outcome.
It may be interesting to include the number of patients who did not experience any adverse events during the course of the treatment in the Safety Profile subsection.
The data presented in figure 1 should be presented in table format and should include the number of patients in each category in addition to percentage of patients.
Please use McNemar’s test to compare the paired nominal data shown in Table 3.
The phrasing of the sentence on line 232 implies that the treatment regimen with Zadiva® has been concluded and will be permanently relapse free. Re-phrasing the sentence to something along the lines of “Based on the findings of our study, 90.85% of the patients treated with Zadiva® for one year were found to be relapse-free during that year” would remove the implication of a permanent result.
To evaluate the treatment efficacy in a retrospective manner, the authors should consider to include an appropriate control group.
Please revise the typo in line 28. --> study
Author Response
This manuscript investigates the safety and effectiveness of dimethyl fumarate as a treatment for relapsing-remitting multiple sclerosis over the course of one year in a patient population in Iran in a retrospective cohort study. This manuscript highlights important safety data and suggests an even more gradually increasing dosing regimen to reduce the number of adverse effects. There are several issues that warrant the editor’s attention listed below.
Several authors have affiliations with a pharmaceutical company, including one author with their only affiliation listed as the same pharmaceutical company. Please clarify whether there is any conflict of interest or if this pharmaceutical company is involved with Zadiva® or NanoAlvand Co.
Further clarification would be appreciated with regard to the Author Contributions, where V.G. is listed as the sole author involved in the investigation. According to author affiliations, V.G. is affiliated with the Department of Business of a pharmaceutical company as well as with the faculty of veterinary medicine of the Tabriz Branch of the Islamic Azad University.
Please attach the informed consent form as a supplement.
The length of the study period is not included in the Materials and Methods section. There are several references in the statistical analyses in the Results section referring to differences after one year of treatment with Zadiva®, but there is no explicit mention of the study length. Please include this information in either the Treatment Protocol subsection or any other appropriate subsection.
Additionally, the background information does not include any information regarding the expected length of time between relapses in patients not treated with Zadiva® at any point in the paper. This information would be important to make statements regarding the efficacy of the medication as one of the claims of the paper is a significant reduction in relapses (Lines 37, 164, and 271).
Please include the breakdown of male and female patients in the Study Population subsection within the Materials and Methods section and discuss critically how this might have impacted the results.
Please provide additional explanation for the rationale supporting the decision to utilize a different treatment regimen (120mg OD to 120mg BID to 120mg OD + 240mg OD to 240mg BID) from the treatment regimen in the medication insert of Zadiva® (which is the same as the standard treatment regimen for generic dimethyl fumarate).
The Results section includes a table of previous DMTs patients had received prior to Zavida® initiation, but there is no mention of whether these patients stopped their previous DMT prior to beginning treatment with Zavida®, and if so, how long they stopped prior to initiating the Zavida® treatment. Please clarify and discuss how this might have impacted the therapeutical outcome.
It may be interesting to include the number of patients who did not experience any adverse events during the course of the treatment in the Safety Profile subsection.
The data presented in figure 1 should be presented in table format and should include the number of patients in each category in addition to percentage of patients.
Please use McNemar’s test to compare the paired nominal data shown in Table 3.
The phrasing of the sentence on line 232 implies that the treatment regimen with Zadiva® has been concluded and will be permanently relapse free. Re-phrasing the sentence to something along the lines of “Based on the findings of our study, 90.85% of the patients treated with Zadiva® for one year were found to be relapse-free during that year” would remove the implication of a permanent result.
To evaluate the treatment efficacy in a retrospective manner, the authors should consider to include an appropriate control group. |
Thank you so much for your constructive comments. We have carefully revised the manuscript accordingly.
We appreciate your attention to detail and agree that transparency in the affiliations and contributions of the authors is paramount. To clarify, two of our co-authors, who were affiliated with the pharmaceutical company, have been removed from the author list due to potential conflicts of interest related to this project. We have retained only the primary individual from the medical and research department of the pharmaceutical company who was partly involved in the design of this study. Furthermore, please be assured that the pharmaceutical company's role was limited to providing funding for the publication of the study. They were not involved in the collection, analysis, or interpretation of the data, which was conducted independently by the academic authors to maintain scientific integrity and avoid potential biases. The Author Contributions section has been revised for better clarity and congruence. We hope this addresses your concerns, and we apologize for any confusion caused in the initial submission. We are committed to maintaining transparency and integrity in our research.
Thank you for your suggestion. We have included the informed consent form as a supplementary document, as you requested
Thank you for your insightful observation. we've now updated the manuscript to properly document the details with regard to the duration of treatment with Zadiva®. To be clear, this is an observational study, and at the time of analysis, all patients included had been administered Zadiva® treatment for at least one year. The safety evaluation, discontinuation rate, and patient response to the treatment were all assessed during this first year. I apologize for any confusion that may have initially arisen due to the omission of this information.
Thank you for bringing up these points. As a real-world study, our focus was primarily on evaluating the treatment outcomes in a single group of patients receiving DMF. Comparisons were made primarily with the patients' pre-treatment data as a reference point. To support the claim of a significant reduction in relapses, we relied on the comparison between the pre-treatment data and the relapse data observed during the study period. By assessing the change in relapse rates within the same patient group before and after treatment, we aimed to demonstrate the treatment's efficacy in reducing the occurrence of relapses in this specific context.
We have now included this breakdown in the Study Population subsection of the Materials and Methods section as per your advice. Based on your suggestion, we have calculated the female-male ratio for different SOC categories. Our analysis revealed a dominance of females in all SOC categories (see table 2.).
Thank you for your comment. In clinical trials like DEFINE and CONFIRM, the standard treatment regimen for DMF was indeed used. However, in our real-world study conducted in the outpatient clinic at Amir A'lam Hospital, we initially implemented the gradual treatment regimen based on the recommendation to improve gastrointestinal and flushing tolerability outside of the study environment, in order to maximize the benefits for the majority of our patients. As a real-world study, our methods aimed to reflect the clinical practices and challenges faced in real-world settings, and thus, we chose to adopt a modified treatment approach to enhance patient tolerability and adherence to DMF. We have revised the manuscript accordingly.
Thank you for highlighting this important point. In our study, prior to initiating treatment with DMF, all patients were indeed on other DMTs which include interferon beta-1a (IM and SC), interferon beta-1b (SC), glatiramer acetate, fingolimod, and teriflunomide. To comply with standard clinical practice and minimize potential interactions or carryover effects, there was a washout period in which previous DMTs were discontinued before the commencement of DMF treatment. The length of the washout period varied depending on the specific DMT previously administered and its half-life. Patients transitioning from interferon therapies or glatiramer acetate typically do not require a washout period due to their relatively short half-lives. However, for patients switching from fingolimod or teriflunomide, a washout period of approximately 4 weeks is typically recommended due to the longer half-life of these medications.
As per your recommendation, we have now included the number of patients who did not experience any adverse events during the course of the treatment in the Safety Profile subsection. Specifically, out of the 142 patients who were administered the treatment, 81 did not report any adverse events
The information in Figure 1 has been revised and transformed into a table format (Table 3) that includes both the number and percentage of patients.
Thank you for your suggestion. In response to your feedback, we have applied McNemar’s test to compare the paired nominal data shown in Table 3. This statistical test is more appropriate for our dataset as it is designed to analyze paired nominal data and test for changes in proportions between two related groups, in this case, our pre- and post-treatment groups.
Based on the McNemar’s test, we found a highly significant difference in the number of relapses before and after treatment with Zadiva® DMF (p-value < 2.2e-16). The odds ratio of 9.92 indicates that the odds of patients having a relapse after the treatment are approximately 10 times higher than before the treatment. The 95% confidence interval for the odds ratio is between 5.60 and 19.14, suggesting a high degree of certainty about this estimate.
These findings have been included in the updated Table 4 and we believe they offer more robust and accurate statistical insights. We appreciate your guidance in helping us to strengthen our data analysis.
We agree with your observation that the original phrasing could imply a permanent outcome, which was not our intention. We have revised the sentence as per your suggestion to clarify that the relapse-free status pertains to the one-year treatment period with DMF (Zadiva®).
Thank you for your thoughtful suggestion about including a control group. While we agree that control groups are a cornerstone of many research designs, we decided against it in this particular study for a few reasons:
1-Nature of the Study: This is a real-world, observational, retrospective study. Our intention was to analyze the effects of DMF (Zadiva) on patients with RRMS who have already been treated with it, based on real-world data, rather than a traditional, prospective controlled trial setting.
2-Comparison to Pre-treatment Baseline: In our study, we used each patient's condition prior to DMF (Zadiva) treatment as a baseline for comparison. Thus, each patient effectively serves as their own control, which can be a powerful form of analysis in a real-world setting.
We understand the inherent limitations of our study design and acknowledge them in our discussion. |

Round 2
Reviewer 2 Report
We acknowledge the significant improvements and clarifications by the authors and have no additional comments.